

# Validation of a camera-based intra-hour irradiance nowcasting model using synthetic cloud data

Philipp Gregor[1], Tobias Zinner[1], Fabian Jakub[1], and Bernhard Mayer[1]

[1]Ludwig-Maximilians-Universität München, Munich, Germany

**Correspondence:** P. Gregor (philipp.gregor@lmu.de)

**Abstract.** This work introduces the novel short-term nowcasting model MACIN, which predicts direct normal irradiance (DNI) for solar energy applications based on hemispheric sky images from two all-sky imagers (ASI). With a synthetic setup based on simulated cloud scenes, the model and its components were validated in depth. We trained a convolutional neural network to identify clouds in ASI images and derive their height and motion using sparse matching. In contrast to other studies, all derived

cloud information from both ASIs and multiple timesteps are combined into an optimal model state using techniques from data assimilation. This state is advected to predict future cloud positions and compute DNI for lead times up to 20 minutes. For the cloudmasks derived from the ASI images we found a pixel accuracy of $94.66\%$ compared to the references available in the synthetic setup. The relative error of derived cloud base heights is $4\%$ and cloud motion error is in the range of $0.1\mathrm{ms}^{-1}$. For the DNI nowcasts, we found an improvement over persistence for lead times larger than one minute. Using the synthetic

setup, we computed a DNI reference for a point and also an area of $500\mathrm{m} \times 500\mathrm{m}$. Errors for area nowcasts as required, e.g., for photovoltaics plants, are smaller compared to errors for point nowcasts. Overall, the novel ASI nowcasting model and its components proved to work within the synthetic setup.

## 1 Introduction

Clouds are a major modulator of atmospheric radiative transfer, showcased by their ability to shadow the ground. This influence

on the irradiance impacts the production of renewable energy through photovoltaic (PV) and concentrating solar power (CSP) plants. These fluctuations in produced power are a limitation for the usability of PV power. Unexpected variations in power production poses a challenge for the integration into power grids (Katiraei and Agüero, 2011). Prior knowledge of upcoming fluctuations and therefore short-term irradiance prediction can help mitigate this drawback of PV power production (West et al., 2014; Boudreault et al., 2018; Law et al., 2016; Chen et al., 2022; Samu et al., 2021; Saleh et al., 2018, e.g.,).

Especially knowledge of future direct irradiance is important for solar energy applications as it can be blocked completely by clouds within seconds to minutes. Multiple models for intra-hour direct normal irradiance (DNI) nowcasting have been developed to predict this variability of direct irradiance. Many of these rely on so called all-sky imagers (ASI), ground based cameras capturing hemispheric sky images (e.g., Peng et al., 2015; Schmidt et al., 2016; Kazantzidis et al., 2017; Nouri et al., 2022). The general idea is to extract cloud information from these images, predict future cloud positions and accordingly

estimate irradiance for the next minutes. The applicability of low cost consumer grade cameras makes setups with multiple ASIs



financially feasible, and increasing sizes of installed PV plants also require more measurement positions to expand nowcasted areas. The eye2sky (Blum et al., 2021) network showcases the widespread use of multiple ASIs for regional coverage and nowcasting.

Common tasks for ASI based DNI nowcasting are the extraction of cloud position and motion. Li et al. (2011) established a method for the classification of pixels based on color values and thresholding. A similar method exploiting a library of reference clearsky images was introduced to consider different atmospheric conditions and background variations for the large field of view of ASI (Shields et al., 2009; Chow et al., 2011; Schmidt et al., 2016). Also convolutional neural networks (CNN) have been proven to work beneficially for these tasks (Ye et al., 2017; Dev et al., 2019; Xie et al., 2020; Hasenbalg et al., 2020) when trained on densely labeled data. Fabel et al. (2022) demonstrated the use of a CNN to distinguish not only clear and cloudy pixels, but further separated clouds into three subclasses for low, mid and high layer clouds. Further on, Blum et al. (2022) projected cloudmasks of multiple imagers onto a common plane and combined them for an analysis of spatial variations of irradiances. Exceeding the focus on cloudmasks, Masuda et al. (2019) combined a camera model and synthetic images of LES cloud fields to derive fields of cloud optical depth from images.

Setups with multiple ASIs allow for the estimation of cloud base height using stereography (Nguyen and Kleissl, 2014; Beekmans et al., 2016; Kuhn et al., 2018b). Nouri et al. (2018) used four ASIs to derive height information and even a 3-dimensional cloud representation for irradiance nowcasting. Three ASIs were used by Rodríguez-Benítez et al. (2021) for three independent DNI nowcasts which are finally averaged into a mean DNI nowcast.

Whilst measurements of irradiance through pyranometers are point measurements, nowcasting methods are usually targeted at solar power plants and therefore receiver areas. Kuhn et al. (2017b) compared nowcasts against area irradiance values derived using a camera monitoring shadows on the ground in combination with point irradiance measurements (Kuhn et al., 2017a). They found improvements compared to persistence for situations with high irradiance variability. ASI nowcasts for eight pyranometer measurement sites distributed over roughly $1\mathrm{km}^2$ were computed in Nouri et al. (2022). This study found reduced errors if nowcasts and measurements were averaged over all sites before error calculation in comparison to errors of individual point nowcasts.

Apart from application on real world images, Kurtz et al. (2017) applied a DNI nowcasting model to synthetic ASI images of cloud scenes from large eddy simulation (LES) models. These images were generated using a 3D radiative transfer model. This synthetic application comes with the advantage of optimal knowledge of the atmospheric state, which was used to showcase the problems introduced by the viewing geometry of ASIs.

In this study we introduce the novel model for all-sky image based cloud and direct irradiance nowcasting (MACIN) and use synthetic data to validate and apply it. This DNI nowcasting model is based on a setup with two ASIs. We use state of the art techniques, e.g., to derive cloudmasks using a CNN which was trained on sparsely labeled data. Further on, cloud base height (CBH) is derived by stereography and cloud motion by sparse matching. This derived data is fed into a horizontal model grid using a method inspired by data assimilation, which has similarities to the cloudmask combination of Blum et al. (2022) but allows to use images of multiple timesteps and is used for nowcasting future states and not just analyzing the current situation. Predicted cloudiness states are projected to the ground and converted into DNI. We apply the techniques to synthetic



ASI images generated from simulated LES cloud fields. This allows us to validate the derived quantities as well as the overall nowcast performance. Further on, this allows for in depth validation of DNI nowcasts not just for single point measurements but also for areas as it is important for PV and CSP plants. Sect. 2 describes the synthetic data used throughout the study, the methods to derive information from ASI images as well as the MACIN nowcasting model. Additionally, the quantities

and metrics used for validation are explained. Sect. 3 describes the validation of derived cloudmasks, cloud base height, and cloud motion as well as the full DNI nowcasting model. The results of the validation are analyzed and discussed to affirm the presented methods and explain error sources. Conclusions can be found in Sect. 4 as well as a brief description of possible follow-up work.

## 2 Methods

In this section the basics about the data and methods used is described. This includes an explanation of the synthetic data and all-sky images as well as the methods used to derive information about clouds in Sect. 2.1. The DNI nowcasting model which utilizes this information is outlined in Sect. 2.2. Additionally, reference quantities and metrics for validation are given in Sect. 2.3.

### 2.1 Synthetic data and all-sky images

The synthetic data has been prepared by Jakub and Gregor (2022). This dataset is a 6h LES run computed with the cloud model UCLA-LES (Stevens et al., 2005). The horizontal resolution is 25m and LES output fields are given every 10s. The initial atmospheric profile was chosen to produce a single shallow convection cloud layer with cloud base height of roughly 1000m developing from cloud fraction of 0% in the beginning to roughly 100% at the end of the simulation after 6h. For more details and impressions of the cloud scenes used in this study, the reader is referred to Jakub and Gregor (2022).

This dataset provides realistic cloud situations and the possibility for detailed benchmarking. Primarily, the variable cloud liquid water content ($lwc$) is used. To calculate optical properties of clouds, the effective radius is needed. Since the LES output field does not contain this information, a fixed number density of $120 \cdot 10^6 \mathrm{m}^{-3}$ was assumed and effective radius of cloud droplets was calculated following Bugliaro et al. (2011). For simplicity, other atmospheric parameters like water vapor, temperature, pressure and molecular composition from the LES output are neglected within this study and the US Standard

Atmosphere (Anderson et al., 1986) is assumed. While these atmospheric parameters and their variations are in general not negligible for radiative transfer, the setup for this study was simplified to focus on clouds as a major modulator of irradiance. Within this study, the sun was assumed to be at a constant zenith angle of $30°$ to the south.

For synthetic images generated from these LES cloud fields we assume a fisheye camera model corresponding to the OpenCV fisheye module Bradski (2000). The parameters for this projection model were derived by calibration of a CMS Schreder ASI-

16 camera. This ASI features a $180°$ FOV fisheye objective to capture hemispheric images of the cloud situation. Within this study, we use two different approaches to generate all-sky images from LES cloud scenes. For both approaches, we generate images with the viewing geometry derived according to the fisheye camera model for an ASI-16 imager. As our methods are





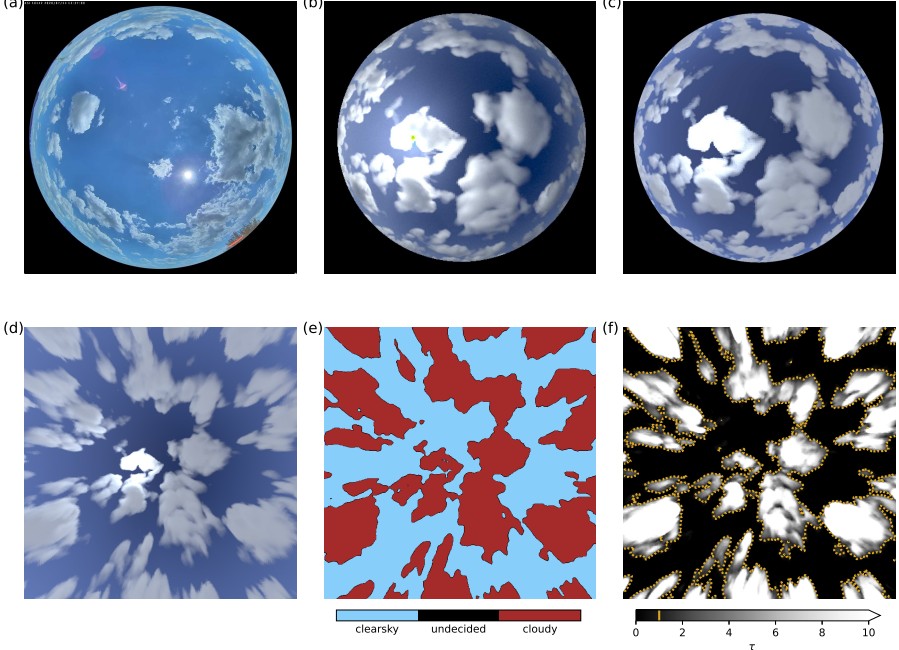

**Figure 1.** (a) Real ASI image captured with a CMS Schreder ASI-16 in the Bavarian countryside (48°10'50.3"N, 11°00'27.4"E) on 14 July 2020. (b-d) Synthetic images for LES time 9900s generated using (b) MYSTIC, (c) ray-marching and (d) ray-marching followed by projection. (e) Cloudmask derived from the projected ray-marching image and (f) LES cloud optical depth $\tau$ in line of sight with additional yellow contour illustrating $\tau_{\mathrm{thresh}} = 1.0$. Only few pixels are labelled *undecided* by the CNN as depicted in (e).

developed to work with cameras, which are not necessarily calibrated spectrally, the images are only roughly optimized to resemble colors of the ASI-16. We use a simple spectral camera model with whitebalance, black level, gamma correction and an upper intensity limit to convert radiances into pixel values.

One of the image generation methods uses synthetic radiances from the Monte Carlo 3D radiative transfer model MYSTIC (Mayer, 2009), which does not introduce any simplifying assumptions in radiative transfer. These radiances can be converted into synthetic images using the camera model. While MYSTIC radiances are physically correct, they are computationally expensive. Computation of these radiances for a single image require multiple cpu-hours and therefore this approach was used for 29 images with a resolution of $240 \times 240$ pixel only. In contrast, our second approach is only a rough approximation of radiative transfer. We use a ray-marching technology commonly applied in the computer gaming industry (eg; Schneider, 2018; Hillaire, 2016) to trace through volumetric media. For every pixel, many small steps along the line of sight are marched through the atmosphere. At every step, the in-scattered light into the line of sight of the simulated imager is computed using local optical properties of the atmosphere. This is summed up to compute the overall light reaching the simulated imager. Schneider (2018) computes at each step the direct radiation from the sun to estimate the amount of light scattered into the direction of the imager. With this approach, multiple scattering is only roughly parametrized although it may be dominant in regions of high



cloud optical thickness. Therefore, we use the original marching together with a different method to calculate the amount of in-scattered light. Direct and diffuse irradiances are calculated with a two-stream radiative transfer model (Kylling et al., 1995) on tilted independent columns of the LES cloud field. For each ray-marching step, the local irradiances are used to estimate the amount of direct and diffuse light scattered towards the simulated imager. This technique is implemented using the OpenGL framework and allows us to generate images of $960 \times 960$ pixel within seconds. Generated images are interpolated to the original ASI resolution in a postprocessing step for both generation methods. Figures 1a-c show a real world image and images generated using MYSTIC and ray-marching. Because of the low computational cost of image generation, we work with ray-marching images throughout this work if not stated otherwise. We derived cloudmasks from both MYSTIC and ray-marching images to confirm the usability of the latter for our purpose.

As a first step in working with the generated images, the camera model is applied to project them onto a horizontal ground parallel image plane. During this reprojection, image features may be distorted and blurred. However, it allows to work on a projected image plane parallel to the ground simplifying further image processing. Figures 1c and 1d display an image as captured by the ASI and its projected correspondence as generated using ray-marching. While the original ASI resolution is of $1920 \times 1920$ pixel, we project images to $480 \times 480$ pixel for use within our nowcasting model.

### 2.1.1 Cloudmasks

The most important information to obtain from all-sky images is the classification of pixels as cloudy or clear. Convolutional neural networks (CNN), which are commonly applied for image segmentation have also been applied to images of clouds to generate cloudmasks (e.g., Dev et al., 2019; Xie et al., 2020; Fabel et al., 2022). Also our cloudmask derivation relies on CNN, we used the DeeplabV3+ network structure (Chen et al., 2018). The set up and training of the CNN is outlined briefly in the following, a more detailed explanation can be found in Appendix A1 as well as an description of how the training data was labeled by hand. Training was done using 793 hand labeled projected images from the ASI-16 depicting various cloud situations. Segmentation classes are *cloudy*, *clear* and *undecided*. As the definition of *cloudy* and *clear* areas in images is often hard even for human observers, the CNN training is designed to ignore *undecided* image regions. The CNN is set up to reproduce the *cloudy* and *clear* labeled regions, but is free to fill in regions labelled as *undecided* by hand without impact on the training. The CNN thus fills in regions for which the cloud state is ambiguous or indistinctable to humans based on the definition of *cloudy* and *clear* for regions, where this is obvious for humans. From the CNN, we obtain cloudmasks as a segmentation of an ASI image into the classes *cloudy*, *undecided* and *clear* with respective scalar values of 1.0, 0.5 and 0.0. Figures 1d and 1e give a synthetic image and the derived cloudmask. For comparison, the LES cloud optical depth ($\tau$) traced in line of sight for each pixel is given in Fig. 1f.

### 2.1.2 Cloud base height from stereo matching

In order to map cloudmasks to 3-dimensional coordinates, cloud base height (CBH) is required. For the experiments presented here, two ASIs are located in a 500m north-south distance. I.e., for each timestep two viewing angles can be exploited to derive CBH. Features from simultaneous ASI images of the same cloud scene are sparsely matched using efficient coarse to fine





patchmatch (CPM; Hu et al., 2016), a pixel based pyramidal matching method. For a grid of pixels on the first input images, DAISY feature descriptors (Tola et al., 2010) are computed and their best matching counterparts in the second image are determined. As a result, we obtain a list of matched pixels from both images, which are supposed to depict the same part of a cloud. We use the derived cloudmasks to filter matched pixels, these must be marked as *cloudy* in the corresponding cloudmasks for both images to be accepted. Using the known camera geometry, a cloud base height can be derived for each matched pair

of pixels with the mis-pointing method developed by Kölling et al. (2019). This results in up to multiple thousand feature positions per pair of simultaneously captured images which theoretically allows for a fine grained treatment of CBH. However, the nowcasting model presented in this study currently assumes a single cloud layer. Therefore an image wide average CBH is derived from the mean height of the feature positions.

### 2.1.3 Cloud motion

To predict future shading by clouds, cloud motion needs to be derived. Using the CPM matching algorithm on consecutive images taken in intervals of 60s, we obtain matches describing the displacement of features. The computed cloudmasks are used again to exclude matches lying outside of detected cloud areas. The average image cloud base height and the camera model is used to scale the detected pixel movement to physical velocities within the assumed plane of clouds. A dense cloud motion field is obtained by nearest neighbor interpolation of these sparse velocities.

## 2.2 Nowcasting model

The nowcasting model presented in the following uses the derived cloudmasks, CBH and cloud motion to predict future cloud situations and corresponding irradiance estimates. Therefore, cloudmasks and cloud motion are represented as variables on a horizontal 2-dimensional grid, which will be referred to as cloudiness state and velocities. The 2-dimensional grid and all input data is assumed to be on one ground parallel horizontal level at the height given by the derived CBH, multi-level clouds

are not yet represented as such by our model. Using a simple advection scheme, future states of these 2-dimensional fields are predicted. From these future cloudiness states, irradiance estimates are computed. To exploit all derived cloudmasks and cloud motions, these are combined into an optimized initial state of the horizontal 2-dimensional fields. The nowcasting model therefore consists of three major parts: a simple advection method, a method inspired by data assimilation to determine the initial state and a radiative transfer parametrization to calculate DNI from the cloudiness state. These three parts are explained

more closely in the following.

### 2.2.1 Advection scheme

The nowcasting model is based on a 2-dimensional grid with grid spacing of $\Delta x = \Delta y = 10\mathrm{m}$ and number of grid points $N = M = 1600$ in $x$- and $y$-direction respectively, therefore covering 16km×16km. Variables on each grid point are cloudiness state cm and cloud velocities in $x$- and $y$-direction, $u$ and $v$. Starting from an initial state at the first iteration $t_0 = 0\mathrm{s}$ and a





temporal resolution of $\Delta t = 60$s, future cloudiness states at times $t_i = t_0 + i \cdot \Delta t$ are computed using advection as

$$\mathrm{cm}_{t_{i+1}}(n,m) = \mathrm{cm}_{t_i}(\tilde{n}, \tilde{m}), \tag{1}$$

$$\tilde{n} = n - int(\lambda \cdot u(n,m)), \tag{2}$$

$$\tilde{m} = m - int(\lambda \cdot v(n,m)), \tag{3}$$

where $\lambda = \Delta t / \Delta x$. As the grid points require discrete coordinates, the coordinates $(\tilde{n}, \tilde{m})$ determined by advection using

physical velocities are restricted to integers. This constrains actually representable velocities to multiples of $\Delta x / \Delta t$. Continuous boundary conditions are assumed. The same advection scheme is applied to the horizontal velocities fields $u_t(n,m)$ and $v_t(n,m)$ as well.

### 2.2.2  Data Assimilation

Cloudmask and horizontal velocity field from one imager at a single time together with an estimation of cloud base height

would be sufficient to initialize the advection model. However, for each nowcast we do have cloudmasks and velocities from two imagers with different viewing geometries and multiple timesteps. In order to make use of as much information as possible for the initial state, we therefore employ a method similar to 4D-var data assimilation (Le Dimet and Talagrand, 1986) in numerical weather prediction models. The general idea is to define a scalar function of an initial model state, which measures differences of model states and measurements. This so called cost function is then iteratively minimized to find an optimal

model state for the given measurements and cost function.

Of course, the difference of model state and measurements has to be minimized at matching times. Therefore, model states for multiple time steps are computed from the initial state at time $t_0$ using the previously described advection $M$. Model cloudiness states at time $t_k$ will be denoted as $\mathrm{cm}(t_k) = M(\mathrm{cm}, t_k)$ with the initial cloudiness state cm. Horizontal velocities $u$ and $v$ are described analogously. We define the cost function $J$ for $L$ timesteps in the interval $[t_0, t_l]$ and two ASIs ($p \in 1, 2$)

as

$$
\begin{aligned}
J(\mathrm{cm}, u, v) = \sum_{N,M} \sum_{l=0}^{L} \sum_{p=1}^{2} \ & \left( \frac{1}{\sigma_{\mathrm{cm}}} \cdot (\mathrm{cm}(t_l) - \mathrm{cm}_{\mathrm{meas},l,p})^2 \right. \\
& + \frac{1}{\sigma_{uv}} \cdot (u(t_l) - u_{\mathrm{meas},l,p})^2 \\
& \left. + \frac{1}{\sigma_{uv}} \cdot (v(t_l) - v_{\mathrm{meas},l,p})^2 \right) \\
& + R(u,v)
\end{aligned}
\tag{4}
$$

with measurements of cloudmasks $\mathrm{cm}_{\mathrm{meas},k,l}$ and horizontal velocities at timestep $k$ from imager $l$ interpolated to the model grid. For better readability, the summation over all grid points is indicated by $\sum_{N,M}$. The coefficients $\sigma_{\mathrm{cm}} = 0.1$ and $\sigma_{uv} = 10.0 \mathrm{ms}^{-1}$ are supposed to account for uncertainties in the respective measurements but are mainly used as tuning parameters here. More complex, non-scalar coefficients could differentiate e.g. for varying measurement quality within ASI images or between different imagers but require characterization of the system, which is usually not available. The additional regularization





term denoted as $R(u,v)$ is used to suppress measurement errors, especially outliers in the velocity field as

$$R_{uv}(u,v) = \sigma_{R,uv} \cdot \left( (\nabla u)^2 + (\nabla v)^2 \right) \tag{5}$$

The tuning parameter $\sigma_{R,uv} = 250\mathrm{s}^{-1}$ was chosen to smooth the velocity field. As cloudmasks are especially hard to derive from ASI images in the bright region of the sun, measurement values are excluded from assimilation, if they are derived from an image region of $2.5°$ around the sun. This excludes not only erroneous cloudmask values derived for the bright sun but also zero velocities derived from the static sun position. Due to the limited complexity of the advection scheme and the high resolution observations from images, a background state is not used. This means successive nowcast runs are independent as states from previous model runs for the nowcast start time are not considered in additional terms in Equ. **??**. The cost function is minimized using the bounded L-BFGS-B algorithm (Zhu et al., 1997). For efficient optimization, the advection model and cost function were implemented using the PyTorch framework (Paszke et al., 2019), which allows for automatic calculation of the adjoint of the cost function. The optimized model state is finally used for the actual nowcast as initial state of the advection model.

### 2.2.3 Radiative transfer parametrization

Direct solar irradiance is reduced by interaction with molecules, aerosol and clouds. For this study, we assume that short term changes in direct irradiance are mainly caused by clouds and neglect other variations. DNI is parametrized using previous irradiance measurements on site together with predicted cloudmasks. The idea is to derive references for occluded and non-occluded cases from the measurements. Depending on the cloudiness state, the DNI is then interpolated from these references. Therefore, a time series of clearsky index values (CSI) $k$ is constructed from DNI measurements as the ratio of measurements and a simulated clearsky $\mathrm{DNI}_{\mathrm{clear}}$. From this time series, values for $k$ are extracted for two sub-series for occluded ($k > 0.9$) and non-occluded ($k < 0.1$) times respectively. We define the occluded CSI $k_{\mathrm{occl}}$ and non-occluded CSI $k_{\mathrm{clear}}$ as the exponentially weighted mean with a half life time of $10\mathrm{min}$ from the respective measurement subsets. CSI values for a non occluded and a fully occluded sun are interpolated linearly. Therefore, a sun disk of $0.5°$ opening angle and at the given sun elevation and azimuth is projected onto the 2-dimensional model grid. The mean cloudiness state of all grid points in the sun disk ($\mathrm{cm}_{\mathrm{sun}}$) is used to calculate DNI for time $t$ as

$$\mathrm{DNI}(t) = \mathrm{DNI}_{\mathrm{clear}} \cdot \left( (1 - \mathrm{cm}_{\mathrm{sun}}(t)) \cdot k_{\mathrm{clear}} + \mathrm{cm}_{\mathrm{sun}}(t) \cdot k_{\mathrm{occl}} \right) \tag{6}$$

The exponentially weighted mean is used for the computation of $k_{\mathrm{occl}}$ and $k_{\mathrm{clear}}$ in order to smooth latest fluctuations and provide a values for all times.

### 2.3 Synthetic data experiment setup

The synthetic setup allows us to compare quantities derived by our nowcasting model to synthetic reference values. Within this study we simulate a measurement setup as shown in Fig. 2. Images are rendered with MYSTIC and ray-marching as explained





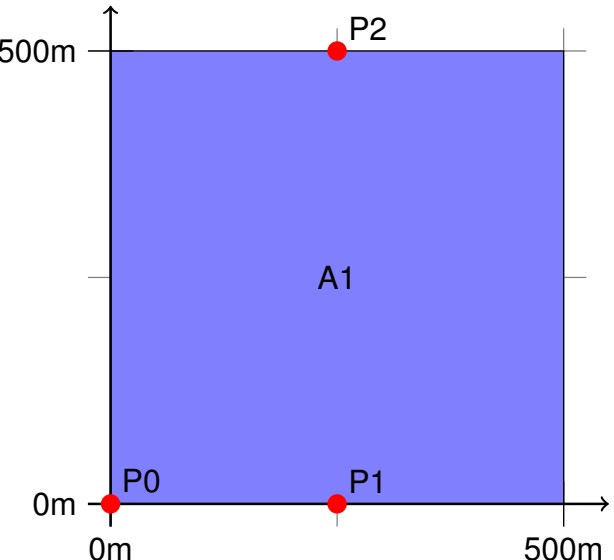

**Figure 2.** Measurement setup used within the synthetic data experiments with images for points P0, P1, P2 as well as DNI references for point P1 and area A1

in Sect. 2.1 for a synthetic ASI at P0 to compare both methods. For the actual nowcasting and all other applications in this study, ray-marching images for P1 and P2 are used.

The validation quantities used within the experiments in Sect. 3 are explained in the following. To validate derived cloud-masks, the cloud optical depth ($\tau$) is traced in line of sight for every pixel of the corresponding ASI image. By applying a threshold to the resulting $\tau$-fields we can calculate reference cloudmasks. Fig. 1f shows an example $\tau$-field. These are used for the validation of our derived cloudmasks. The reference cloud base height is computed to comply with the the view of an ASI. The origin of information and in this case cloud height as seen from below is determined by the last scattering before reaching the ASI sensor using further capabilities of MYSTIC beyond the computation of radiances. A reference for cloud motion is hard to define as clouds in the LES simulation – as in nature – are not moving as solid objects, but may change size and shape or even appear and disappear. Even wind velocities at cloud level may therefore not be an exact benchmark for the overall observable cloud motion. Within this study we therefore use vertically integrated liquid water path (lwp) from the LES fields as an indicator for horizontal cloud distribution. The maximum cross-correlation between domain wide lwp of two successive timesteps is assumed to be a reference for average cloud motion. This reference describes the mean displacement for all timesteps of the LES cloud data. However, clouds are convectively reshaping, growing and shrinking in this data, which makes this cloud motion definition vague. The synthetic data allows for a more direct validation of cloud motion by freezing an LES cloud field for a timestep and shifting its position. This basically simulates scenes of pure advection without any convective effects. To implement this advective case for cloud motion validation, we use two images of the same cloud scene but taken from different positions. The choice of the assumed time difference between the images $\Delta t$ defines the advective cloud





velocity. For simplicity, we use images taken in $500\text{m}$ north-south distance as represented by P1 and P2. Assuming $\Delta t = 60\text{s}$ we obtain theoretical cloud velocities of $-8.3\text{ms}^{-1}$ and $0\text{ms}^{-1}$ meridionally and zonally.

The Monte Carlo 3D RTE solver MYSTIC was used to compute not just radiances for images but also the true direct normal irradiances at the ground. We calculated direct normal irradiance for two different synthetic references as depicted in Fig. 2. A DNI point reference is simulated at P1, and an area reference of the $500\text{m} \times 500\text{m}$ region A1 with the ASIs centered at the northern and southern boundary at P1 and P2. As a benchmark for the DNI nowcasting model, persistence nowcasts for start time $t_0$ and nowcast time $t$ are calculated from measured $\text{DNI}_{\text{meas}}$ as

$$\text{DNI}_{\text{pers}}(t) = \text{DNI}_{\text{meas}}(t_0). \tag{7}$$

The metrics used for validation are

$$\text{RMSE} = \sqrt{\frac{1}{N} \sum_{i=1}^{N} (x_i - x_{\text{ref},i})^2}, \tag{8}$$

$$\text{NRMSE} = \sqrt{\frac{1}{N} \sum_{i=1}^{N} \left(\frac{x_i - x_{\text{ref},i}}{x_{\text{ref},i}}\right)^2} \tag{9}$$

and

$$\text{MBE} = \frac{1}{N} \sum_{i=1}^{N} (x_i - x_{\text{ref},i}) \tag{10}$$

with the quantity to be evaluated $x$, its corresponding reference $x_{\text{ref}}$ and the number of values $N$. Additionally, we use the pixel accuracy

$$\text{PA} = \frac{\text{CCLD} + \text{CCLR}}{N_{px}} \tag{11}$$

with the number of correctly *cloudy* or *clear* classified pixels CCLD and CCLR respectively as well as the overall number of pixels $N_{px}$.

## 3    Validation using synthetic data

### 3.1    Cloudmasks

Our CNN cloudmask model was successfully trained and validated on hand labeled real world images as explained in Appendix

A1. Through the evaluation of cloudmasks we aim to show that it is reasonable to apply our cloudmask CNN to the synthetic images of this study. We calculated path cloud optical depth ($\tau$) for all viewing angles of our ASI and every desired timestep. Together with a threshold this gives a reference cloudmask. To validate pixelwise cloud classifications, we use a threshold of $\tau_{\text{thresh}} = 1.0$ to create reference cloudmasks from $\tau$. Values of $\tau \leq \tau_{\text{thresh}}$ are linked to *cloudy* areas in this $\tau_{\text{thresh}}$-cloudmasks.



**Table 1.** Contingency table for cloudmask classes from CNN and cloud optical depth $\tau$ in line of sight thresholded by $\tau_{\mathrm{thresh}} = 1$ as reference. All values are given in %.

|  |  | reference | | |
|---|---|---|---|---|
|  |  | $\tau < 1.0$ | $\tau \geq 1.0$ | $\sum$ |
|  | *clear* | 46.43 | 2.54 | 48.97 |
| CNN | *undecided* | 0.22 | 0.25 | 0.47 |
|  | *cloudy* | 2.33 | 48.23 | 50.56 |
|  | $\sum$ | 48.98 | 51.02 | 100 |

We evaluated CNN cloudmasks from ray-marching images for position P1 and 360 timesteps in $60\mathrm{s}$ intervals covering all LES times. The contingency table 1 displays the distribution of classes of $\tau_{\mathrm{thresh}}$-cloudmasks against our CNN cloudmasks. In general we find very good compliance. Each of the classes *cloudy* and *clear* makes up about $50\%$ of the compared pixels which corresponds well to the $\tau_{\mathrm{thresh}}$-cloudmasks. Cloudmasks of our CNN exhibit a slight bias towards classifying too less pixel as *cloudy*. The pixel accuracy is $\mathrm{PA} = 94.66\%$ against the $\tau_{\mathrm{thresh}}$-cloudmasks.

Beyond the ray-marching images, we calculated 29 MYSTIC images and computed CNN cloudmasks for these. By doing the same with corresponding ray-marching images, we could ensure that the derived cloudmasks exhibit similar performance for both image generation approaches. As MYSTIC images are physically correct, we conclude that the usage of approximated ray-marching images does not affect the validity of our results.

### 3.2 Cloud base height

We used data from for the entire LES scene and effectively 319 timesteps with clouds for the validation of derived CBH. Ray-marching images taken at P1 and P2 were used to derive CBH as in our nowcasting model and computed scattering positions give the reference CBH. As our nowcasting model assumes a single cloud base height, we average derived CBH per image pair. Fig. 3 shows the derived average CBH and the standard deviation per image pair. For these averaged heights, we obtain a MBE for our miss-pointing method of $50.7\mathrm{m}$, RMSE of $56.9\mathrm{m}$ and NRMSE of $4.0\%$. When subtracting the found bias of $50.7\mathrm{m}$ from the derived image average cloud base heights, RMSE could be reduced to $25.6\mathrm{m}$ and NRMSE to $2.6\%$. However, Fig. 3 shows increasing systematic error up to about reference CBH of $1400\mathrm{m}$. As indicated by the whiskers, not only image wide averages but also the distribution of derived cloud base height for pixels within single images shows good agreement with the reference. The found errors could result from the limited resolution of images and therefore discrete viewing directions, the projection process and the discrete stepping of the ray-marching algorithm. As the errors are in the range or even lower of what other studies find for their derived cloud base heights (Nguyen and Kleissl, 2014; Kuhn et al., 2019; Blum et al., 2021, e.g.,), error sources were not investigated further. Equally, no additional work was done to mitigate the observed systematic errors for use in nowcasting.



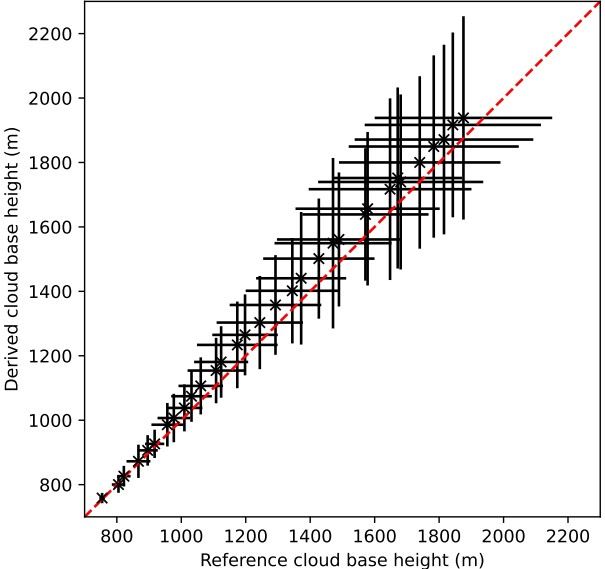

**Figure 3.** Comparison of scene averages of derived and reference cloud base height. Whiskers give standard deviation over pixels for these averages. For better visibility, only every $10th$ timestep and therefore 32 datapoints are shown.

### 3.3 Cloud motion

As wind is not necessarily an exact benchmark for cloud motion in convective cloud scenes, we chose two ways to validate our derived cloud motion for two cases. Firstly, the cloud motion according to the LES is used as a convective case where clouds
also develop and decay. Additionally, we are interested in the performance for pure advection when cloud motion is only the displacement of frozen cloud fields. The advective case allows to derive an exact reference for cloud motion and the convective case allows to validate the quality of the derived cloud motion in case of clouds which change their size and shape.

The validation of cloud motion in the convective case is done on images every $60s$ for LES times from $0s$ to $21540s$. Figure 4 shows the cloud fraction as a function of LES time. As a reference, average displacement of vertically integrated liquid
water path ($\mathrm{lwp}$) between timesteps is calculated using the maximum cross-correlation. This describes mean translation and is thereby a proxy for domain averaged reference cloud motion. Our derived cloud motion vectors are averaged per timestep and for each ASI and compared against this reference. Figure 4 shows zonal and meridional winds derived for both ASI and the reference determined by $\mathrm{lwp}$ cross-correlation. The cloud fraction derived from cloudmasks of an ASI at P1 is given as an indicator of the cloud situation. Up to LES time of approximately $3600s$, no significant visually detectable clouds are present
and therefore no velocities derived. During the period up to approximately $6000s$, derived velocities are relatively unstable over time with changes in estimated velocities of up to $1.7\mathrm{ms}^{-1}$ over $60s$. We relate this to the rapidly changing nature of small convective clouds in combination with low cloud fraction. During this time, some of the small clouds appear and disappear





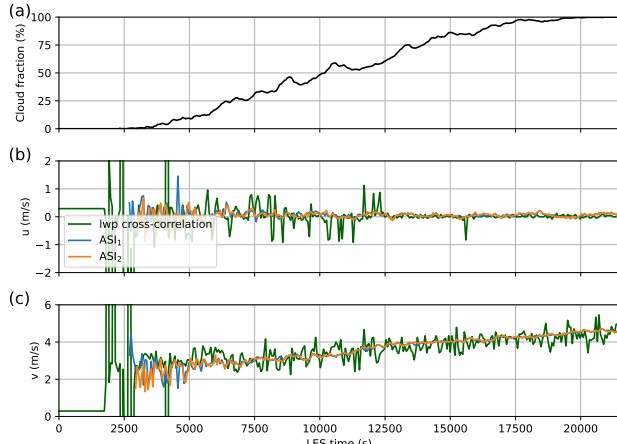

**Figure 4.** (a) cloud fraction from cloudmasks of ASI at P1 for LES times. Per timestep scene averaged cloud motion derived using cross-correlation of the $\mathrm{lwp}$-field of the LES simulation and our cloud motion derivation based on feature matching for east west motion $u$ (b) and south north motion $v$ (c).

in between timesteps and are therefore mismatched. After approximately $6000\mathrm{s}$, derived zonal velocities vary in a range of $\pm 0.5\mathrm{ms}^{-1}$ between timesteps. As there is no initial zonal wind in the LES, zonal cloud motion close to zero matches our

expectations. Meridional velocities increase from about $3\mathrm{ms}^{-1}$ at $6000\mathrm{s}$ to a maximum of $4.7\mathrm{ms}^{-1}$. In general, our derived zonal velocities show a less noisy estimate compared to the reference. The derived velocities from both ASIs show very similar patterns. This further affirms the stability of the cloud motion derivation. However, we do not have an absolute reference to benchmark derived velocities in the convective case as the pure displacement of convective clouds is hard to capture and may differ strongly from main winds. Additionally, we validate derived cloud velocities using artificially advected cloud fields to

overcome this limitation. The same LES times as in the convective validation are used, but each timestep is assumed to be independent. Cloud motion is generated by freezing the cloud field and shifting it for each timestep to obtain an objective reference cloud motion. A shift of $500\mathrm{m}$ from north to south at a time difference of $60\mathrm{s}$ gives a theoretical $u$ of $0\mathrm{ms}^{-1}$ and $v$ of $-8.3\mathrm{ms}^{-1}$. No velocities were derived in the absence of clouds up to approximately $2500s$. Afterwards, the derived velocities match the theoretical displacements well with RMSE of $0.019\mathrm{ms}^{-1}$ zonally and $0.11\mathrm{ms}^{-1}$ meridionally.

Overall, these results prove the derived cloud motion reliable for the cloud situations used in this study. This can also be seen as a further validation of the derived CBHs as they are necessary for the calculation of physical velocities.

### 3.4 DNI Nowcasts

The evaluation of the nowcasting model is done in multiple steps which are described and discussed in the following. First, MACIN is compared against persistence to evaluate overall performance. Additionally, variations of MACIN using ideal cloud-

masks were run to investigate the implications of errors in the CNN derived cloudmasks. These variations will be called *cloud-*





*mask variation* and *continuous cloudmask variation* hereafter. Finally, a simplification of MACIN is used to assess possible benefits of the expensive assimilation of MACIN. This variation will be referred to as *simple variation*. For MACIN and all its variations, one nowcast run was started every 60s for LES times 60s to 21540s, which resulted in 359 nowcasts runs. The maximum nowcast lead time was chosen as 20min. Nowcasts timesteps exceeding the maximum LES time of 21600s are
discarded. DNI nowcasts are always derived simultaneously for the point P1 and the area A1. In the following, error values are given for point DNI and in brackets for area DNI if not stated otherwise.

Figures 5a-b show average RMSE and MBE for the point nowcasts of persistence, MACIN and *cloudmask variation* groupped by lead time. Figures 5c-d give the same for area nowcasts. The errors of persistence and MACIN give the over-all performance of the introduced nowcasting model and are therefore analyzed first. Persistence nowcasts start without error
at lead time 0min, but RMSE increases strongly up to approximately constant $300\mathrm{Wm}^{-2}$ ($250\mathrm{Wm}^{-2}$) after 6min. Persistence MBE increases linearly up to approximately $50\mathrm{Wm}^{-2}$ linked to the tendency of growing cloud fraction over time. MACIN exhibits non-zero RMSE at nowcast start, but a smaller increase of RMSE over time than persistence. Therefore, MACIN outperforms persistence already for lead times larger than 1min in terms of RMSE. Typical improvement over persistence for these longer lead times is thereby on the order of $50\mathrm{Wm}^{-2}$ ($50\mathrm{Wm}^{-2}$) and more. In general, the RMSE of nowcasts for
areas is about $50\mathrm{Wm}^{-2}$ lower compared to nowcasts for points. MBE is mostly negative for MACIN with magnitudes in the range of persistence MBE. The non-zero RMSE at lead time 0min may be a result of erroneus cloudmasks in the region of the sun, errors in the RT parametrization or smearing out during the assimilation because of multiple time steps and viewing geometries.

To further investigate this initial nowcast error, a *cloudmask variation* of MACIN was run. Instead of cloudmasks from the
CNN, perfect cloudmasks were used as input for the nowcasting model. These perfect cloudmasks are derived from the LES cloud optical depths in line of sight $\tau$ (see also Sect. 3.1) with a threshold of $\tau_{\mathrm{thresh}} = 1.0$ to distinguish between *cloudy* and *clearsky*. By using these perfect cloudmasks for nowcasting, influence of cloudmask errors within the nowcasting model can be assessed. As for the persistence and MACIN, nowcast errors for the *cloudmask variation* are given in Fig. 5. In general, the RMSE of the *cloudmask variation* is very similar to the RMSE of MACIN. This suggests that the CNN cloudmasks
provide a good estimate of the cloud situation for our nowcasting. However, for lead time 0min the *cloudmask variation* outperforms MACIN by $31\mathrm{Wm}^{-2}$ ($32\mathrm{Wm}^{-2}$) and converges to the RMSE of MACIN for lead times of 3min and more. The point MBE of the *cloudmask variation* is in the beginning about $0\mathrm{Wm}^{-2}$, the negative MBE of MACIN especially during the first minutes of the nowcasts can therefore be associated to erroneous cloudmasks in the vicinity of the sun. The small improvement for larger lead times when using perfect cloudmasks might also be a result of the convectively growing, shrinking
and reshaping clouds. As the nowcasting model cannot describe these processes, perfectly outlining clouds in the beginning may not be that relevant for longer lead times. Again, the non-zero RMSE of the *cloudmask variation* for lead time 0min may result from errors in the RT parametrization or smearing out by assimilation. To further investigate the implications of the RT parametrization, the *continuous cloudmask variation* was run, again differing from MACIN only by the used cloudmasks. The RT parametrization maps model cloudiness states linearly to DNI values. The model cloud states rely on cloudmasks
with discrete values for the three classes (*clearsky*, *cloudy*, *undecided*) while actual cloud optical depth and therefore DNI





are continuous. To check, whether this discrete representation causes significant parts of the error, continuous cloudmasks are used. These are derived from $\tau$ used for the cloudmask validation, but comply to the exponential attenuation of intensity in radiative transfer by $\mathrm{cm_{cont}} = 1 - \exp(\tau)$. The *continuous cloudmask variation* uses these continuous cloudmasks. Resulting errors of the *continuous cloudmask variation* are not depicted as they strongly resemble errors of the *cloudmask variation* with
slight improvements of RMSE in the range of about $5\mathrm{Wm^{-2}}$. Therefore we conclude that the RT parametrization and discrete nature of cloudmasks is not a major error source and the non-zero RMSE for lead time $0\mathrm{min}$ is a result of the smearing out during the assimilation.

A further variation of MACIN was run to assess benefits of the assimilation scheme. Therefore, the *simple variation* of MACIN was run with just a single cloudmask and velocity field from the ASI at P1 as input. The sun region is not masked
out in cloudmask and velocitity field for the *simple variation*. With this variation we would like to assess possible benefits by the additional complexity and computational cost of MACIN. Resulting errors differ from errors of MACIN mainly for point nowcasts. For lead time $0\mathrm{min}$ the RMSE of the *simple variation* is about $300\mathrm{Wm^{-2}}$. For larger lead times, the RMSE resembles the RMSE of MACIN but is approximately $75\mathrm{Wm^{-2}}$ larger. The MBE of the *simple variation* is strongly negative with values around $75\mathrm{Wm^{-2}}$ and even more for lead time $0\mathrm{min}$. As the sun region is not masked out in the *simple variation*
and the cloudmask CNN tends to classify the sun in synthetic images as *cloudy*, the initial model cloud state and therefore DNI for lead time $0\mathrm{min}$ results in large errors. In case of clearsky the erroneously *cloudy* detected sun is steady and therefore this "cloud" does not move and gives an offset for all lead times. This explains the large RMSE offset and the large negative MBE. We are aware, that these larger errors are mainly due to the co-location of ASI and nowcasted point in our setup. Still, this demonstrates the capabilities of our nowcasting model to use multiple data sources to reduce errors. E.g. when using projected
images of ASIs at different positions and superimpose one over the other for derived CBH, the sun is in different regions of the images. When we exclude per ASI the immediate region of the sun from the used cloudmask, cloudmask information from another ASI is used to fill in this region. Thus, erroneous cloudmasks in the region of the sun can be mitigated by the assimilation. Additionally to the RMSE presented here, the mean absolute error was computed as an additional metric for the nowcasts. It is not shown here as the mean absolute error in general resembles the structure of the RMSE.

## 390    4    Conclusions

In this study we introduced the novel all-sky imager (ASI) based direct normal irradiance (DNI) nowcasting model MACIN, which adapts ideas of 4D-Var data assimilation. We validated MACIN against synthetic data from LES cloud scenes. The nowcasting model is designed to consider measurement setups with multiple ASI and derive point and area nowcasts of DNI. Therefore, we derive cloudmasks, cloud base height and cloud motion from ASI images and combine these into an initial cloud-
mask state for a 2-dimensional horizontal advection model. Predicted cloudmasks are projected to the ground and converted to DNI using previous DNI measurements.

Cloud scenes from a shallow cumulus cloud field computed using UCLA-LES (Stevens et al., 2005) with cloud fraction between $0\%$ and $100\%$ were used for validation and in-depth analysis of the nowcasting system and its components. For these





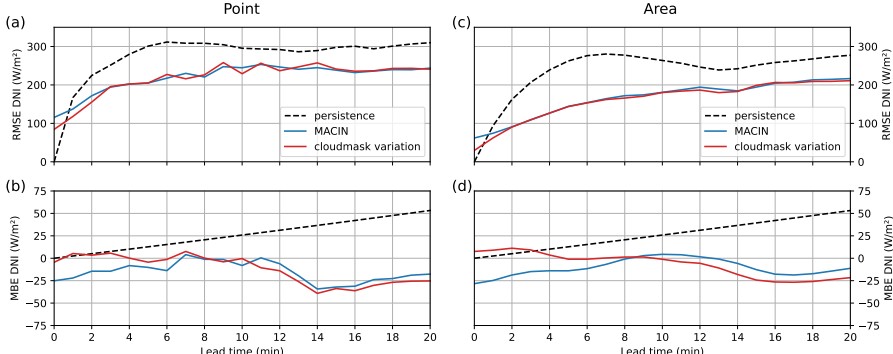

**Figure 5.** (a) RMSE and (b) MBE for 359 point DNI nowcasts compared to point measurements evaluated per lead time. Nowcasts were done using MACIN and the *cloudmask variation*. (c) and (d) show corresponding error values for area nowcasts and DNI area reference.

cloud scenes, synthetic ASI images were generated. DNI at the ground was calculated for synthetic point and area measure-
ments. Reference for cloud optical depth and cloud base height were derived for ASI by tracing through the cloud scenes. With
this data, we validated the cloud detection method relying on a CNN, the cloud base height derivation from stereography and
cloud motion derivation from sparse feature matching of consecutive images. The synthetic setup facilitated a comparison of
DNI nowcasts from MACIN against point and area measurements usually unavailable from observations. Thereby we could
confirm previous findings of a RMSE reduction by spatial aggregation for nowcasts by Kuhn et al. (2018a). Overall, we find
improvements over persistence. In general, the errors correspond to errors found for other ASI based nowcasting systems in
literature (e.g., Peng et al., 2015; Schmidt et al., 2016; Nouri et al., 2022). MACIN gives non-zero errors for point nowcasts
from the start as also observed in e.g. Schmidt et al. (2016) and Peng et al. (2015). Deriving reference cloudmasks from LES
cloud optical depth allowed for an attribution of the initial errors of MACIN to imperfect cloudmasks in the vicinity of the sun,
imperfect DNI estimation and a smoothing of the initial state by assimilation. For applications where these initial errors are
crucial, they could easily be reduced by using persistence nowcasts for small lead times and nowcasts of MACIN for larger
lead times as suggested by Nouri et al. (2022). We did not address this further as it is unlikely to be relevant for operational use,
given that an immediate computation of DNI nowcast, transfer to consumers and reaction of their system seems unrealistic. By
comparing further simplified nowcasts relying only on a single imager we demonstrate the capability of the nowcasting model
to make beneficial use of multiple ASIs and the assimilation scheme.

A limitation of this study is the restricted set of 360min of cloud data and single sun zenith angle. Future work will apply
the nowcasting model to real world data to consider manifold cloud scenes and sun positions. This step is necessary to further
confirm the benefits of the model. Additionally, we plan to use the synthetic setup for in-depth investigation of theoretical error
sources of ASI nowcasts, e.g. to investigate errors introduced by using advection to predict future cloud states and neglecting
convective development of clouds.





*Data availability.* The LES cloud data used for this study is published (Jakub and Gregor, 2022) and can be obtained at https://opendata. physik.lmu.de/5d0k9-q2n86. For further data, e.g., the full syntetic image sets or reference DNIs please contact the authors of this study.

## Appendix A

### A1   Cloudmask CNN

Convolutional neural networks (CNN) are used frequently in image segmentation tasks. The wide variety of possible atmo-
spheric conditions, ligth situations and cloud types even within single ASI images allows only for limited success with classical e.g. color and threshold based methods for cloudmask derivation. E.g. Dev et al. (2019) demonstrated the possibility to use CNNs for segmentation of all-sky images, Fabel et al. (2022) even demonstrates a segmentation into different classes of clouds.

For the application of a single layer advection model, we aim at a segmentation between cloudy and clear areas of an image. A major piece of work is the generation of training images. As the nowcasting model and CNN is intended for real world
applications beyond this study, 793 ASI images were hand labeled and split into a train and validation dataset of 635 and 158 images respectively. These are normalized using the channel-wise mean and standard deviation over all training images. All images of both datasets were scaled to $512 \times 512$ pixel. For training, random excerpts of $256 \times 256$ pixel were cropped and randomly mirrored or rotated by $90°$ to artificially increase the amount of training data by augmentation. Hand labeling was done through a tool we designed for this task, which subdivided a randomly chosen and projected ASI training image
into so called superpixels (Achanta et al., 2012), continuous regions with similar color information and limited distance. Each superpixel can be assigned on of the three classes *cloudy*, *clear* or *undecided*. The subdivision into superpixels allows for faster labeling of pixels belonging together. The labeling tool allows for the selection of the number of superpixels, therefore also small regions may be labeled precisely. As clouds and clearsky are not always precisely distinguishable and their definition based on visual appearance is hard, we also offered the label *undecided*. This label marks regions which are hard or cumbersome
for humans to classify and therefore left out. In the training of the neural network these *undecided* pixels are considered as such, i.e. the CNN is not challenged to label these regions according to potentially miss-labeled training data but may learn more from regions where humans are sure about the proper label. Example images from the validation set, corresponding hand labeled segmentation and CNN segmentation are shown in Fig. A1. The CNN and its training is described in the following. We chose the DeeplabV3+ (Chen et al., 2018) CNN architecture which is designed using an Encoder-Decoder structure as also
the UNet (Ronneberger et al., 2015) used by Fabel et al. (2022) does. For the encoder, we use a ResNet34 (He et al., 2015) pre-trained on the Imagenet dataset (Russakovsky et al., 2014). Three output channels were chosen associated to the three classes. Training was done using the Adam optimizer (Kingma and Ba, 2014) with a custom sparse-soft-cross-entropy loss (ssce). This ssce actively ignores pixels which are labeled as *undecided* in the ground truth and only focuses on *cloudy* and *clear* pixels in the ground truth. This is done using

$$y_{\mathrm{mask},i,j} = 1 - y_{\mathrm{gt},i,j,\mathrm{undecided}} \tag{A1}$$



$$\text{LogSoftmax}(y_{i,j,c}) = \log \left( \frac{\exp(y_{i,j,c})}{\sum_d \exp(y_{i,j,d})} \right) \tag{A2}$$

$$\text{ssce}_{i,j} = \sum_{c \in \{\text{cloudy,clear}\}} \text{LogSoftmax}(y_{\text{pr},i,j,c}) \cdot y_{\text{gt},i,j,c} \cdot y_{\text{mask},i,j} \tag{A3}$$

where $y_{\text{pr},i,j,c}$ is the predicted value for the $i$-th pixel of the $j$-th training image and the class $c$. Correspondingly, $y_{\text{gt},i,j,c} \in \{0,1\}$ is the ground truth value. While ssce is necessary for optimization, this loss is meant to give mainly intermediate scores of performance of the segmentation CNN. Therefore, also a metric called mean intersection-over-union ($mIoU$) is used in a sparse version as

$$\text{I} = \sum_{i,j} \sum_{c \in \{\text{cloudy,clear}\}} y_{\text{gt},i,jc} \cdot y_{\text{pr},i,j,c} \cdot y_{\text{mask},i,j} \tag{A4}$$

$$\text{U} = \sum_{i,j} \sum_{c \in \{\text{cloudy,clear}\}} (y_{\text{gt},i,j,c} + y_{\text{pr},i,j,c}) \cdot y_{\text{mask},i,j} - \text{I} \tag{A5}$$

$$\text{mIoU} = \frac{\text{I}}{\text{U} + \epsilon} \tag{A6}$$

with $\epsilon = 10^{-7}$ for numerical stability. This metric is designed to represent a ratio between correctly classified pixels in comparison to overall classified pixels, again adapted by us to ignore *undecided* ground truth pixels. It was computed after every epoch on the entire validation dataset. We used a batch size of 26 images and a learning rate of $7 \times 10^{-5}$. After 48 epochs of training, mIoU $= 0.968$ was reached for the CNN as used within this study. For the prediction of cloudmasks, the label of a pixel is derived from the output channel with maximum value. This is mapped to scalar values as 0 for *clear*, 1 for *cloudy* and 0.5 for *undecided* to obtain the final CNN cloudmasks.

*Author contributions.* PG developed the nowcasting model, performed the validation and wrote the manuscript in its current form. FJ computed the LES cloud fields and supported their integration into this study. TZ and BM assisted the model development and validation and contributed to the manuscript. TZ and BM prepared the proposal for the BMEL project.

*Competing interests.* At least one of the (co-)authors is a member of the editorial board of Atmospheric Measurement Techniques. The peer-review process was guided by an independent editor, and the authors also have no other competing interests to declare.

*Acknowledgements.* This work was supported by the German Federal Ministry of Food and Agriculture (BMEL) through the FNR (Fachagentur Nachwachsende Rohstoffe e.V.) project NETFLEX-LMU FKZ22400318. We are grateful to Josef Schreder for the support and



professional advice in context of all-sky imagers, especially the CMS Schreder ASI-16 imagers. Additionally, Josef Schreder supported this work on behalf of the CMS - ING. DR. SCHREDER GMBH by providing the ASI-16 all-sky imagers. Images of two ASI-16 were used in the development of the cloudmask algorithm. The synthetic ASI used throughout this studied was modelled to resemble the geometry of an
ASI-16.



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



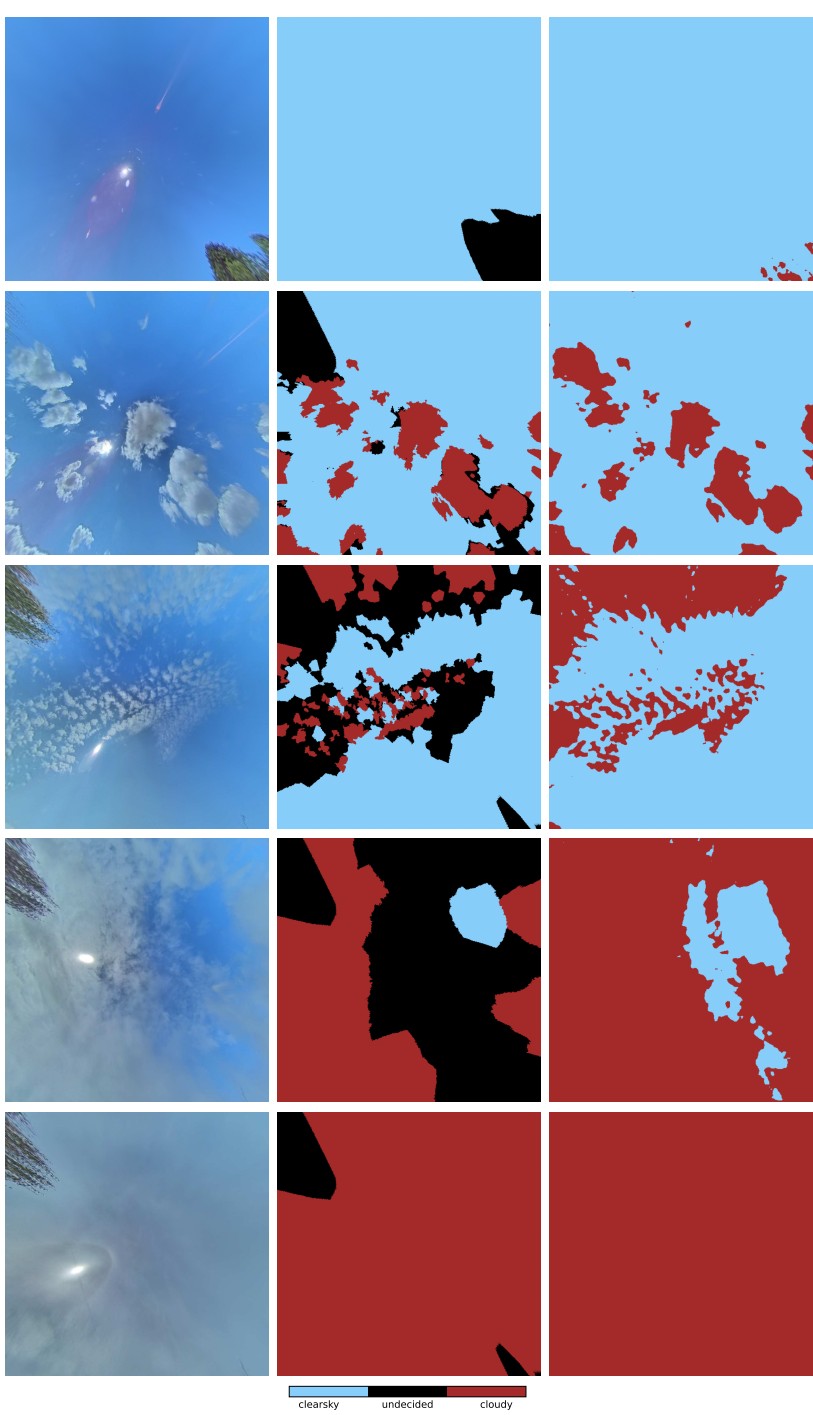

**Figure A1.** Example images from the validation set (left) hand labeled segmentation (middle) and cloudmask predicted by the trained CNN (right).