# Peer review of "Validation of a camera-based intra-hour irradiance nowcasting model using synthetic cloud data"

_Atmospheric Measurement Techniques, 2023_

## Author Comment (AC1)

**Response to review comment 1**

**Manuscript:** AMT-2023-26

**Title:** Validation of a camera-based intra-hour irradiance nowcasting model using synthetic cloud data

**Authors:** Philipp Gregor, Tobias Zinner, Fabian Jakub, and Bernhard Mayer

We thank the anonymous referee for the comments on the manuscript and suggestions to improve its quality. These are addressed in the following. The authors' answers are printed in blue. A version of the manuscript with tracked changes is provided along the updated manuscript.

**Summary:**

This study introduces a short-term nowcasting model that combines techniques such as machine learning and data assimilation in a novel fashion to help in predicting the direct normal irradiance. Validation of the models and methods is thorough and the authors take the time to explain the interpretation of their results. The addition of an "undecided" class to the training step is clever, especially when tied to their loss function which works to sidestep uncertainties in classification. Data assimilation from two separate imagers is used appropriately and adds an extra layer of context for the initial state. Overall, the paper advances the state-of-the-art of nowcasting by combining several innovative methods and could serve as a baseline for future research in the radiation or energy communities using such techniques.

**Specific Comments:**

1. The caption for Fig. 2 needs to be more descriptive. As well, Fig. 2 is not adequately described in the main text when it is first referenced although lines 251-253 do add more context. I would strongly suggest providing the reader with that context for the figure to start with by adding more information in the caption.
   For easier interpretation of Figure 2, we added the axis description (x, y) and extended the caption. Additionally, the main text was extended to introduce P0, P1, P2 and A1 along the reference of Fig. 2.

2. In Sect. 3.2, it would be helpful if a histogram of the retrieved cloud base height is added. This would allow a reader to quantify the performance of the derived base height for the entire set of scenes without limiting samples as has been done for Fig. 3.
   We changed Fig. 3. to two histograms of (a) height of matched pixels and (b) image average cloud base height to include all samples. The text in Sect. 3.2 was adapted accordingly.

3. Starting from line 340, the authors use a value within parentheses when describing the irradiance. It is unclear as to what these values are referring to, particularly as there is a preceding value before the parentheses as well. For instance, lines 343 - 344 say "Typical improvement over persistence for these longer lead times is thereby on the order of $50\mathrm{Wm}^{-2}$ ($50\mathrm{Wm}^{-2}$) and more" but both values being the same creates confusion. I would recommend

introducing the parameter within the parentheses first or explaining it at the top of the paragraph.

The first value gives the error for point DNI whereas the value in the brackets gives the error for area DNI. We tried to indicate this in line 335-336 (previous version, now 348-349): *In the following, error values are given for point DNI and in brackets for area DNI if not stated otherwise.*

For further clarification we extended this to: *Errors for point and area forecasts show similar characteristics. Therefore, it is discussed jointly in the following. If not stated otherwise, error values are given for point DNI and in brackets for area DNI.*

4. There are a number of grammatical errors overall that will need to be corrected before publication. For instance, in lines 20-21, the phrasing should be "Since direct irradiance can be blocked completely by clouds within seconds to minutes, knowledge of future direct irradiances is especially important for solar energy applications.". The incorrect use of adverbs and articles in many places interrupts the flow and might particularly detract a reader from the point of the sentence or a paragraph which is why I am adding this issue as a major comment.

   We rechecked the manuscript completely and tried to correct grammatical errors and improve overall readability. We are thankful for your comment and hope that we were able to improve the manuscript.

5. In the appendix, it is mentioned that the ResNet encoder of the CNN uses pre-trained weights from ImageNet. This seems like an unnecessary step as the ImageNet classes are oriented at natural object detection (and not clouds) and transfer learning from a pre-trained ResNet would not necessarily reduce the convergence time on a task such as cloud detection. Could the authors clarify why a pre-trained model is better as opposed to training from scratch for this application?

   The ImageNet dataset is not focussed on cloud or sky images and does not feature according classes. Therefore we agree that transfer-learning from a pre-trained ResNet encoder may seem unnecessary. In our tests, however, we found the pre-trained weights to be helpful for faster convergence during training on the cloud segmentation data. Unfortunately we have not looked deeply into this difference in convergence. Arguing based on an intuitive understanding, we suggest that the reduced convergence time is due to the fact that especially the weights of the first convolution layers are pre-trained to focus on gradients in the input images. Although cloud boundaries are often fuzzy, there still are gradients present especially in the vicinity of cloud borders. Using a ResNet with randomly initialized weights, additional training is necessary for this focus on gradients. Due to the simple availability of ResNet weights pre-trained on ImageNet, we compared training with and without these and found improved convergence with pre-trained weights.

**Technical Corrections:**

1. The LaTeX equations have not rendered correctly in the preprint. For instance, line 207 has a question mark instead of equation numbers. This needs to be corrected.

The broken link to Equ. 4 was fixed in the LaTeX document. Apart from this, we found no other broken links.

2. The cloud optical depth threshold in line 273 should be reversed to say $\tau > \tau_{thresh}$ is classified as a cloudy region.
   We corrected it to $\tau \geq \tau thresh$.

3. In line 388, there is mention of mean absolute error. Since this metric is not presented in the main text or appendix or supplement, I would recommend removing this sentence as it is unnecessary. The RMSE and MBE already provide sufficient quantification.
   The reference to mean absolute error in line 388-389 was removed.

---

## Author Comment (AC2)

**Response to review comment 2**

**Manuscript:** AMT-2023-26

**Title:** Validation of a camera-based intra-hour irradiance nowcasting model using synthetic cloud data

**Authors:** Philipp Gregor, Tobias Zinner, Fabian Jakub, and Bernhard Mayer

We thank the anonymous referee for the comments on the manuscript and suggestions to improve its quality. These are addressed in the following. The authors' answers are printed in blue. A version of the manuscript with tracked changes is provided along the updated manuscript.

**Summary:**

The presented study is a valuable addition to the increasingly important topic of very short-term solar irradiance nowcasts. The methods used in the presented model are a mixture of existing and new approaches.

Here, the data assimilation approach derived from numerical weather prediction for an optimal initial state is very promising. Furthermore, the validation strategy using synthetic ASI images derived from LES simulations should be positively highlighted. This enables the comprehensive validation capabilities as mentioned by the authors.

The manuscript is coherently structured and explains the applied methods sufficiently

I can agree with the comments of the previous referee.

**General comments to improve quality:**

- When reading the manuscript I was sometimes confused about synthetic and real data. E.g. readers might associate "DNI measurements" with instrument (pyrheliometer) measurements. It was also not clear to me if ASI images for training the CNN model were real ASI images. Please take care of a clear distinction when real and synthetic data has been used
  We added a sentence in Sect. 2.1.1 to clearly state the use of real ASI images for the CNN training.  We struggle to get rid of the term "measurement" completely, as e.g. it is a typical term for input data in assimilation and the distinction between model values of the nowcasting model and modelled synthetic images and simulated DNIs is not clear anymore. To avoid confusion, we added remarks to the sections where the term "measurement" is used.

- The decision of choosing DNI vs. GHI or DHI can be explained in more detail. The estimation of spatial distributed diffuse horizontal irradiance (DHI) from ASI is a generally rarely addressed issue in ASI based nowcasting and worth to be mentioned. DHI is also needed for transposition modelling needed for solar energy applications. I can also imagine that the used setup with MYSTIC RT modelling and LES is capable in addressing DHI estimations.

We added additional explanation for the choice of DNI as a subject of investigation instead of GHI or DHI.

We agree, that the estimation of spatially distributed DHI is a challenging aspect of irradiance nowcasting and the used synthetic setup could be applied to validate also DHI estimations. Especially MYSTICs capability of modelling 3D radiative transfer would be valuable therefore.

Spatial and temporal persistence is still state of the art for modelling diffuse irradiance in ASI based cloud modelling contexts (e.g.; Blum et al. 2022; Chow et al. 2011) to our knowledge. Only statistical and machine learning approaches actively model DHI different than persistence. We did not develop any more advanced method yet. As Chow et al. 2011 states, "[T]he primary factor modulating GHI from its clear sky value is the presence of a cloud between the ground location and the sun as this directly attenuates the solar beam irradiance. Variations in diffuse irradiance caused by changing cloud distribution and optical depth are smaller and generally negatively correlated with beam irradiance". We therefore did not evaluate DHI persistence and its effect on GHI estimation and focused in this manuscript on DNI nowcasting, which we directly relate to the cloud situation. For future work and real world photovoltaic applications, DHI of course cannot be ignored and will be considered.

- Chapter 2.2.2: The quality of the used data assimilation can be better explained and is probably of interest for the reader. A graphical example of model, observation and analysis states might be a solution for this.

  We added a schematic figure to illustrate the quality of the used data assimilation. As stated in the manuscript, the previous model state – background in classical DA terms – is not used by us.

- Persistence comparison: in general, by definition, the quality of persistence decreases with cloud variability. The synthetic data set with increasing cloud cover is therefore an ideal basis for an investigation of forecast skill of the model vs. persistence. Therefore, a comparison of forecast skill with respect to cloud fraction would improve the manuscript.

  We initially disregarded comparing forecast skill per cloud fraction or variability class in the maunscript as the number of forecasts is limited (359) and the classes get quite small for the typical separation into 8 classes (e.g.; Nouri 2022). Additionally, the shallow cumulus setup is also a limited sample of possible atmospheric conditions. Below, we attached a Figure of RMSE over lead time calculated for 5 classes of cloud fraction, which is not included in the manuscript. We added a short paragraph at the end of Section 3.4 discussing the results, but also a disclaimer about the limited significance.

  We opted for 5 classes in this Figure as a tradeoff between bin size and number of forecasts per class. Area nowcasts show more stable curves and are therefore chosen here. While differences in overall RMSE and improvement of MACIN over persistence are definitely significant between classes of different cloud fractions, the explanation is straightforward and as you already suggested directly linked to the variability. Medium cloud fractions mean higher variability and overall larger nowcast errors. As persistence is assuming exactly the absence of variability, MACIN shows nowcast improvements especially for conditions of high variability (medium cloud fractions). Also, the absence of multilayer clouds and

constant sun position simplifies this explanation. Altogether, this does not add significant new insights over existing literature in our opinion and we therefore only mention it briefly in the manuscript. For real world applications with a larger number of nowcasts and multilayer clouds and very location dependent weather conditions and distribution of cloud fraction and variability this nevertheless seems like an important part of a thorough investigation and will be considered for future studies.

[Figure]

**Technical comments:**

General: The figures are a quite small and therefore difficult to read. Possibly they can be made larger for the final version.

We agree with the figures being quite small in the document. The figures were sized according to the directives in the LaTeX template, therefore we hesitated to increase the overall figure size. However, the font size in the figures may of course be increased for better readability. For a final version, we will discuss this with the copy editing team.

Line 3: You might add that the CNN was trained on "real ASI images" to avoid confusions.

We adapted the text to clearly indicate the training on real ASI images.

Line 27: It should be Eye2Sky instead of eye2sky

Changed to Eye2Sky.

Line 195: Please check k,l it might be mixed with l,p as defined in the equation above

It was indeed mixed up. We changed the text according to the equation.

Line 330: The definition of cloudmask variation and continuous cloudmask variation might be introduced here. It takes a while to understand the difference

We decided not to include the definition to avoid breaking the overview structure of the first paragraph. We added a hint towards later explanation and expanded the explanation in the corresponding paragraphs to improve understandability of the difference.

Line 338: grouped instead of groupped

Changed to grouped.

Line 347: RT definition not introduced

We added a definition of radiative transfer (RT).

**References**

Blum, N. B., Wilbert, S., Nouri, B., Stührenberg, J., Lezaca Galeano, J. E., Schmidt, T., Heinemann, D., Vogt, T., Kazantzidis, A., and Pitz-Paal, R.: Analyzing Spatial Variations of Cloud Attenuation by a Network of All-Sky Imagers, Remote Sensing, 14, 5685, https://doi.org/10.3390/rs14225685, 2022.

Chow, C. W., Urquhart, B., Lave, M., Dominguez, A., Kleissl, J., Shields, J., and Washom, B.: Intra-hour forecasting with a total sky imager at the UC San Diego solar energy testbed, Solar Energy, 85, 2881–2893, https://doi.org/10.1016/j.solener.2011.08.025, 2011.

Nouri, B., Blum, N., Wilbert, S., and Zarzalejo, L. F.: A Hybrid Solar Irradiance Nowcasting Approach: Combining All Sky Imager Systems and Persistence Irradiance Models for Increased Accuracy, Solar RRL, 6, 1–12, https://doi.org/10.1002/solr.202100442, 2022.